# Suppression of CCT3 Inhibits Tumor Progression by Impairing ATP Production and Cytoplasmic Translation in Lung Adenocarcinoma

**DOI:** 10.3390/ijms23073983

**Published:** 2022-04-02

**Authors:** Shuohua Chen, Yang Tian, Anji Ju, Boya Li, Yan Fu, Yongzhang Luo

**Affiliations:** 1Cancer Biology Laboratory, School of Life Sciences, Tsinghua University, Beijing 100084, China; chen-sh16@mails.tsinghua.edu.cn (S.C.); tianyang2147@gmail.com (Y.T.); jaj16@mails.tsinghua.edu.cn (A.J.); liby15@mails.tsinghua.edu.cn (B.L.); fuyan@tsinghua.edu.cn (Y.F.); 2Beijing Key Laboratory for Protein Therapeutics, Tsinghua University, Beijing 100084, China; 3The National Engineering Research Center for Protein Technology, Tsinghua University, Beijing 100084, China

**Keywords:** CCT3, lung adenocarcinoma, growth, metastasis, ATP production, cytoplasmic translation

## Abstract

Heat shock proteins are highly expressed in various cancers and exert critical functions in tumor progression. However, their expression patterns and functions in lung adenocarcinoma (LUAD) remain largely unknown. We identified that chaperonin-containing T-complex protein-1 subunit 3 (CCT3) was highly expressed in LUAD cells and was positively correlated with LUAD malignancy in the clinical samples. Animal studies showed that silencing CCT3 dramatically inhibited tumor growth and metastasis of LUAD. Proliferation and migration were markedly suppressed in CCT3-deficient LUAD cells. Moreover, the knockdown of CCT3 promoted apoptosis and cell cycle arrest. Mechanistically, the function of glycolysis was significantly inhibited and the total intracellular ATP levels were reduced by at least 25% in CCT3-deficient cells. In addition, the knockdown of CCT3 decreased the protein translation and led to a significant reduction in eukaryotic translation initiation factor 3 (EIF3G) protein, which was identified as a protein that interacts with CCT3. Impaired protein synthesis and cell growth in EIF3G-deficient cells were consistent with those caused by CCT3 knockdown in LUAD cells. Taken together, our study demonstrated in multiple ways that CCT3 is a critical factor for supporting growth and metastasis of LUAD, and for the first time, its roles in maintaining intracellular ATP levels and cytoplasmic translation are reported. Our novel findings provide a potential therapeutic target for lung adenocarcinoma.

## 1. Introduction

Lung cancer is the leading cause of cancer-related morbidity and mortality in the world. There are approximately 2.2 million new patients and 1.79 million deaths worldwide each year, and the five-year survival rate is less than 20% [1]. Lung adenocarcinoma comprises about 40% of all lung cancer cases [2]. There is an urgent need to discover new targets in order to develop effective drugs for LUAD patients.

Heat shock proteins (HSPs) are important chaperone proteins that exert their functions mainly by facilitating protein folding or refolding into the correct conformation and maturation to maintain protein homeostasis [3]. They are critical for cell survival, especially in cancer cells, where continuous synthesis of protein is required to meet the high protein demand for rapid and sustained growth [4]. According to the molecular weights of these proteins, they are usually divided into small HSPs, HSP40, HSP60, HSP70, HSP90 and HSP100 [5]. Some of them are specifically overexpressed in cancer cells to exert a “buffer”-like function for wild-type or mutant client proteins [6]. These client proteins of HSPs participate in plenty of processes involved in cancer development, such as proliferation, apoptosis, angiogenesis and epithelial-mesenchymal transition [7,8]. Based on that, HSPs have emerged as potential targets for developing antitumor agents. Accumulating evidence has demonstrated that HSPs play important roles in tumor progression [9]. However, a systematic investigation of HSPs to identify their importance in LUAD has still not been reported.

Chaperonin-containing T-complex protein-1 (CCT), belonging to HSP60 family group II, is a chaperonin formed by stacking two identical complexes in a back-to-back manner. Each complex is composed of eight different subunits (CCT1-8) [10]. The proteins involved in some essential biological processes, such as the cell cycle and cytoskeleton formation, have been identified in the CCT interactome [11,12,13]. CCT3 is overexpressed in various cancers [14,15,16,17,18,19,20], such as liver cancer, cervical cancer, papillary thyroid carcinoma and gastric cancer. Knockdown of CCT3 induces apoptosis and inhibits tumor proliferation. Although increasing evidence indicates that CCT3 is critical for tumorigenesis and progression, the role of CCT3 in LUAD and the underlying mechanisms are still unknown. 

In this study, to identify the heat shock proteins responsible for LUAD development, we examined HSP gene expression in LUAD tissues and normal tissues using The Cancer Genome Atlas (TCGA) database. Six HSPs were shown to be abnormally expressed in LUAD tissues, and their expression levels were correlated with patients’ survival. Among them, CCT3 was identified as being critical for tumor growth and metastasis of LUAD. By comparing the proteomics changes after CCT3 knockdown, we found depletion of CCT3 significantly influenced the energy supply and protein synthesis pathways. This study demonstrates that CCT3 is an essential protein for LUAD development by regulating multiple functions in cells and provides a potential target for the treatment of LUAD.

## 2. Results

### 2.1. CCT3 Is Up-Regulated in Human LUAD Tissues and Positively Correlated with Tumor Malignancy

To investigate specific HSPs that are responsible for LUAD development, we collected heat shock protein genes (Appendix A) from the Enzo Life Sciences website and compared their transcriptional levels in normal tissues and LUAD tissues with TCGA data. We found 20 genes were abnormally expressed at the transcript level (Figure 1A). Among them, six genes (*CCT3*, *CCT5*, *CCT6A*, *DNAJB4*, *HSPB7* and *HSPD1*) were found to be significantly correlated with patient survival from the high and low expression groups (Figure 1B and Appendix A). 

To further confirm the expression differences of the transcriptome, we compared the relative mRNA levels of the six genes in one human normal epithelial cell line (BEAS-2B) and three LUAD cell lines (A549, H1299 and H460) through RT-qPCR. Results showed that only the transcriptional level of CCT3 had a consistent trend and was significantly up-regulated in all LUAD cell lines compared to the human normal epithelial cell line (Figure 1C and Appendix A). The protein levels of CCT3 were confirmed in these cell lines (Figure 1D).

Intriguingly, we noticed that higher transcriptional levels of CCT3 were correlated with more advanced cancer stages (Figure 1E). Then, the protein levels of CCT3 in lung tissues from healthy individuals and LUAD patients were tested by immunohistochemistry staining. The results showed that CCT3 was significantly overexpressed in LUAD patient specimens and was positively correlated with LUAD malignancy in the clinical samples (Figure 1F,G). Taken together, CCT3 is significantly up-regulated at both transcriptional and translational levels in human LUAD. Meanwhile, all the above results suggest that CCT3 may be a prognostic marker for LUAD.

### 2.2. CCT3 Is Required for Tumor Growth and Metastasis of LUAD

To investigate the role of CCT3 in lung adenocarcinoma growth in vivo, we knocked down CCT3 via lentiviral vectors expressing short hairpin RNA (shRNA) targeting *CCT3* in H1299 and A549 cell lines. Unexpectedly, none of the cells were able to amplify after stable knockdown of CCT3 (data not shown), probably due to the essential function of CCT3 in LUAD cells. To overcome this, we employed a doxycycline-inducible system to induce CCT3 knockdown and confirmed efficient reductions in their mRNA and protein levels after doxycycline treatment (Appendix A).

Then, we subcutaneously inoculated H1299 cell lines harboring inducible shRNAs targeting *CCT3* or non-targeting control into the BalB/C nude mice, respectively. The tumor growth was significantly suppressed upon CCT3 knockdown (Figure 2A). We also noticed that knockdown of CCT3 dramatically reduced the size and weight of the tumor (Figure 2B,C). In addition, the number of Ki67-positive cells was significantly decreased in the CCT3 knockdown group (Appendix A).

Next, we further verified CCT3’s function on tumor growth in an orthotopic model and found significant decreases in total lung weight (Figure 2E) and tumor area (Figure 2D,F) when CCT3 expression was ablated. Additionally, we found that the CCT3 knockdown groups developed less local invasion of tumor cells in comparison with the control groups (Figure 2G). This suggested that knockdown of CCT3 not only inhibited tumor growth but also repressed the metastasis of LUAD. Then, we created a metastatic model by intravenously injecting the tumor cells into mice and harvested their lungs after 60 days. Robust inhibition of lung metastasis was observed in the CCT3-silenced group (Figure 2H,I). Given these observations, our data demonstrate that the knockdown of CCT3 remarkably inhibits both tumor growth and metastasis of LUAD.

### 2.3. Suppression of CCT3 Expression Inhibits Proliferation of LUAD Cells

To explore the mechanism of CCT3 in LUAD progression, we investigated CCT3′s function at the cellular level in vitro. Firstly, we silenced CCT3 with siRNAs and verified the efficiencies at both protein and mRNA levels in H1299 and A549 cell lines (Figure 3A and Appendix A). Then, cell proliferation assays showed that suppression of CCT3 significantly inhibited cell growth of H1299 and A549 cells (Figure 3B). Additionally, the results of the colony formation assay revealed significant reductions in the number of colonies in CCT3-deficient groups compared to the control groups (Figure 3C).

### 2.4. Suppression of CCT3 Expression Induces Apoptosis and Cell Cycle Arrest in LUAD Cells

To investigate the mechanism of cell growth suppression, we employed cell apoptosis analysis using flow cytometry and found a significant increase in annexin-V positive populations in cells with CCT3 depletion as compared to control cells (Figure 3D). In addition, cell cycle distribution was examined, and the results showed that knockdown of CCT3 increased the percentage of LUAD cells in the G1 phase while reducing the population of S-phase cells. We did not observe any population changes in the G2 phase (Figure 3E). Collectively, these findings suggest that the reduced growth of CCT3 knocked down cells is a consequence of the increased apoptosis and cell cycle arrest.

### 2.5. Knockdown of CCT3 Expression Inhibits Migration and Invasion of LUAD Cells

We performed transwell assays to evaluate whether the depletion of CCT3 impaired LUAD metastasis by interfering with the invasion process. The results showed that suppression of CCT3 effectively reduced the number of cells that invaded into the lower chamber compared to the control groups in migration assays (Figure 3F). Similar results were observed in the invasion assays (Appendix A). These data demonstrate that the inhibition of metastasis in the CCT3-knockdown LUAD groups is partly achieved by reducing the abilities of cellular invasion and migration.

### 2.6. Silencing CCT3 Decreases Intracellular ATP Levels by Impairing Glycolysis in LUAD Cells

Since CCT3 is a critical protein in LUAD, we hypothesized that depletion of CCT3 has a broad influence on other aspects of cellular status. To understand the mechanism underlying CCT3’s broad regulation, we investigated whether knockdown of CCT3 has essential influences on proteomic distribution in H1299 and A549 cells. To this end, we first evaluated the protein expression profiles of CCT3 knockdown cells and control cells by TMT-MS (Appendix A). After the abrogation of CCT3 expression, 756 and 997 differentially abundant proteins were identified in H1299 and A549, respectively, with the following cut-off values: protein fold change (siCCT3/siCtrl) > 1.20 or <0.83 with FDR < 0.05. Next, 97 up-regulated and 70 down-regulated proteins were overlapped in H1299 and A549 (Appendix A). 

The pathway enrichment analysis was performed to further understand the biological consequences of those dysregulations [21], and we discovered that those up-regulated proteins in the CCT3 knockdown groups were primarily enriched in pathways related to the mitochondrial matrix and its functions (Figure 4A). Firstly, the morphologies of mitochondria were investigated. The results showed that silencing CCT3 presented elongated mitochondria, whereas cells in the control groups showed round-shaped mitochondria in H1299 and A549 cells (Figure 4B). In agreement with the above observations, the perimeter contour and aspect ratio were increased (Appendix A), whereas circularity and shape factor were decreased in the CCT3 knockdown groups (Figure 4C and Appendix A).

Normally, elongated mitochondria are associated with increased oxidative phosphorylation activity (OXPHOS) to produce more ATP [22]. In our data, GSEA results show that the OXPHOS pathway was notably enriched in the CCT3 knockdown groups (Figure 4D). To our surprise, the intracellular ATP levels were significantly decreased in H1299 and A549 cells after CCT3 knockdown (Figure 4E). Next, the mitochondrial stress test was performed to evaluate the functions of mitochondria (Appendix A). Unexpectedly, only the maximal respiratory capacity was significantly enhanced in the CCT3 knockdown cells (Appendix A). The functions of mitochondria in terms of ATP production and basal respiration were not significantly changed between CCT3 knockdown and control cells (Figure 4F and Appendix A). It indicated that, under the same conditions, the mitochondria produced similar levels of ATP through OXPHOS in CCT3 knockdown and control cells. In tumor cells, aerobic glycolysis is another important process for supplying energy [23]. Therefore, we performed glycolysis stress tests (Figure 4G) and found that silencing CCT3 remarkably decreased the functions of basal glycolysis and glycolytic capacity without affecting the glycolytic reserve function in LUAD cells (Figure 4H,I and Appendix A). Collectively, our findings show that knockdown of CCT3 reduces intracellular ATP levels by inhibiting glycolysis in LUAD cells.

### 2.7. Inhibition of CCT3 Expression Impairs Cytoplasmic Translation in LUAD Cells

In the enrichment analysis of down-regulated proteins, we found the translation factors pathway was significantly enriched (Figure 5A). In addition, the gene set of the cytoplasmic translation pathway was also enriched in the results of GSEA (Appendix A). Hence, we adopted the surface sensing of translation (SUnSET) assay, a method that can monitor the newly synthesized proteins [24], to check the global translation in LUAD cells. We noticed that knockdown of CCT3 led to a significant decrease in protein synthesis (Figure 5B). Further analysis of the proteomics revealed that those proteins down-regulated by CCT3 depletion were dramatically enriched in the translation initiation factor activity pathway (Figure 5C and Appendix A). Among them, we found that the protein levels of EIF3G were decreased with a maximum fold-change after CCT3 knockdown in LUAD cells (Figure 5D and Appendix A).

Since CCT3 belongs to the chaperonin protein family and the mRNA levels of EIF3G were not changed after depletion of CCT3 in LUAD cells (Figure 5E), we hypothesized that CCT3 regulates EIF3G by interacting with it. Then, co-IP assays were performed and showed that CCT3 did have direct physical interaction with EIF3G in LUAD cells (Figure 5F). The results were validated through pulldown assays in vitro (Figure 5G).

In order to explore the effects of EIF3G reduction on global translation and proliferation of LUAD cells, we knocked down EIF3G in H1299 and A549 cell lines using siRNAs (Figure 6A and Appendix A). SUnSET assays showed that knocking down EIF3G significantly reduced intracellular global translation (Figure 6A). Meanwhile, knocking down EIF3G did not change the protein level of CCT3 (Figure 6A). Furthermore, depletion of EIF3G alone significantly inhibited the proliferation and colony formation of LUAD cells (Figure 6B,C), which was the same as the results in the CCT3 knockdown cells. 

Taken together, these data suggest that the inhibitory effect of knocking down CCT3 on proliferation was in part achieved through down-regulating EIF3G—which physically interacts with CCT3—and thereby impairing global translation in LUAD cells.

## 3. Discussion

Many studies have explored the functions of HSPs in tumor progression [25]. However, their expressions in lung adenocarcinoma cells and their corresponding functions are rarely systematically studied. Herein, we identified that CCT3, one of the HSP60 proteins, was highly expressed in LUAD tissues at transcriptional and translational levels compared to the normal tissues and was positively correlated with LUAD malignancy. Furthermore, LUAD patients with higher expression levels of CCT3 possessed shorter survival. These findings agree well with previous research in other cancers [15,16,19], indicating that CCT3 is an important oncoprotein with broad prognostic potential.

In our study, we also demonstrated that depletion of CCT3 significantly inhibits tumor growth and metastasis in vivo and in vitro. However, why and how does CCT3 affect tumor progression? Early studies reported the biological results of CCT3 deficiency in tumor cells were phenotypical, and affected growth, motility, cell cycle and apoptosis [18,20,26]. In this work, we explored these processes in LUAD cells and observed consistent results, which implies that the role of CCT3 in tumor cells is conserved and could be a common target for cancer therapy. It is worth noting that we found the down-regulated proteins in the CCT3 knockdown groups were enriched in pathways of microtubule cytoskeleton organization and G1/S transition (Figure 5A), which supports the results in cell experiments. Specifically, tubulin proteins (TUBB6, TUBA1C and TUBA4A), essential for the formation of the mitotic spindle, were identified as decreased in CCT3 knockdown cells (Appendix A). Tubulins are not only involved in cell cycle progression but also essential for cell movement, since they are important cytoskeleton proteins [27]. Combined with these reports, the present work suggests tubulin deficiency caused by CCT3 knockdown is one of the reasons that depletion of CCT3 inhibits cell motility and induces cell cycle arrest. However, more evidence is needed to support this hypothesis. 

The competition in this research is extremely fierce. Recently, two articles reported the function of CCT3 in lung cancer. Shi et al. demonstrated that CCT3 is up-regulated in non-small cell lung cancer, and ablation of CCT3 has antitumor roles via affection of YAP1 [28]. Xu et al. reported that CCT3 regulates cisplatin resistance of LUAD cells through the JAK2/STAT3 pathway [29]. In contrast to previous reports, we studied the importance of CCT3 in sustaining intracellular energy supply and protein synthesis in LUAD cells. Within cells, ATP, known as energy currency, is the primary source of power used and stored at the cellular level. In general, ATP is mainly produced from glucose metabolism through glycolysis, the tricarboxylic acid cycle (TCA) and OXPHOS. Reducing intracellular ATP levels by about 30% could efficiently inhibit cellular proliferation and even induce apoptosis [30,31,32,33]. In this study, we found the intracellular ATP levels were reduced by at least 25% after knocking down CCT3 in LUAD cells, and glycolytic activity was significantly decreased. The glycolysis process not only provides ATP, but also is the main intracellular source of pyruvate for the TCA cycle to generate ATP through OXPHOS. Therefore, the inhibition of LUAD progression is indicated to be partially due to a lack of adequate ATP supply in CCT3 knockdown cells. Temiz et al. reported that CCT3 suppression changed the levels of intracellular free amino acids [18]. Some of them had reduced intracellular levels, those synthesized from pyruvate and involved in energy metabolism, which helps to reinforce our conclusion regarding the disturbed balance of energy supply in CCT3 deficiency cells. Beyond that, even though we observed the changes in morphology and overexpression of mitochondria proteins, the function of the mitochondria in ATP production remained unchanged in CCT3 knockdown cells. We speculate that mitochondria were regulated in response to a reduction in total intracellular ATP in CCT3-deficient LUAD cells, although this compensation was ultimately unsuccessful. In brief, our results establish a direct relationship between insufficient energy supply and CCT3 deficiency, expanding our understanding of CCT3’s role in LUAD progression. 

Eukaryotic translation, the essential process of creating functional proteins from codons of genetic information, can be mainly divided into initiation, elongation, termination and ribosome recycling [34]. It is critical for proteostasis and cell survival, especially in cancer cells, where continuous synthesis of proteins is required to meet the high protein demand for rapid and sustained growth [4]. Many inhibitors targeting protein translation have been approved as antitumor drugs [35,36]. In this work, we found many down-regulated proteins in CCT3-depletion LUAD cells were enriched in gene sets related to protein synthesis, such as translation factors and mTORC1 signaling pathways (Figure 5A). In agreement with that, the knockdown of CCT3 significantly decreased translation activity in LUAD cells (Figure 5B). Our findings point to a possible mechanism by which CCT3 deficiency inhibits LUAD progression by suppressing protein synthesis.

eIF3, an important translation initiation factor, is comprised of 13 subunits (EIF3A-M) [37]. Our results showed many of the down-regulated proteins involved in the translation factors pathway were subunits of eIF3. Newly synthesized EIF3B, EIF3I and EIF3H have been identified as substrates of the CCT complex, which mediates the correct folding of these three proteins and prevents their degradation [38]. In this study, we identified EIF3G was physically interacting with CCT3 and found EIF3G was decreased with the maximum fold-change among these down-regulated proteins enriched in the translation initiation factor pathway, implying EIF3G is a client protein of the CCT complex. Moreover, knockdown of EIF3G decreased the intracellular protein synthesis, while the protein level of CCT3 did not change (Figure 6A), demonstrating CCT3 is the upstream regulator of EIF3G, and EIF3G only participates in the synthesis of a part of intracellular proteins. In addition, cell growth was inhibited in the EIF3G-deficient LUAD cells. It is consistent with observations in colorectal cancer and bladder cancer, where depletion of EIF3G alone inhibits cell growth [39,40]. What is more, we noticed that decreases in protein synthesis and cell growth caused in EIF3G deficiency LUAD cells were the same as the phenomena observed in the CCT3 knockdown experiments. It indicates that the impaired protein synthesis caused by CCT3 depletion is through down-regulation of EIF3G in LUAD cells. However, EIF3G overexpression could not restore the inhibited proliferation in CCT3-deficient LUAD cells (Appendix A). From the proteomics analysis, we noticed that multiple translation initiation factors were decreased (Figure 5C), including core subunit EIF3B. EIF3G is one of the translation initiation factor proteins that was reduced after CCT3 knockdown. As a result, overexpression of EIF3G alone was not enough to restore the phenotype caused by CCT3 knockdown. In addition, we explored the effect on proliferation of double knockdown of CCT3 and EIF3G, and the results showed that double knockdown of CCT3 and EIF3G was not significantly different from knockdown of CCT3 alone in inhibiting LUAD cell proliferation (Appendix A). This is most likely because the CCT complex assists in folding 10% of all newly synthesized proteins [41], knockdown of CCT3 has a more disastrous effect on intracellular proteostasis, and EIF3G is one of the downstream effector proteins of CCT3. Therefore, the inhibited proliferation of the double knockdown of CCT3 and EIF3G was not significantly different from that of a single knockdown of CCT3 in LUAD cells. Taken together, these data suggest that the inhibitory effect of knocking down CCT3 on proliferation is partly achieved through down-regulating EIF3G, a physical interacting protein of CCT3, to impair global translation in LUAD cells.

In conclusion, our work demonstrates that CCT3 is a key protein for supporting the growth and metastasis of LUAD. Inhibition of CCT3 offers a potential therapeutic approach for the treatment of LUAD. For the first time, we illustrated the underlying mechanism with direct evidence from the perspectives of energy supply and protein translation. That is, the knockdown of CCT3 inhibits the progression of LUAD in part by reducing the production of intracellular ATP and impairing cytoplasmic translation, providing novel mechanisms to explain the critical role of CCT3 in LUAD progression.

## 4. Materials and Methods

### 4.1. Cell Culture, Reagents and Antibodies

Human LUAD cell lines H1299, A549, H460, human normal lung epithelial cell line BEAS-2B and human embryonic kidney cell line 293T were obtained from the China Infrastructure of Cell Line Resources (Beijing, China). Cells were cultured in RPMI1640, DMEM or Ham’s F12 (Wisent, Nanjing, China) supplemented with 1% antibiotic (Wisent, Nanjing, China) and 10% FBS (GIBCO, Grand Island, NY, USA). Puromycin (631305) was purchased from Clontech (Takara Bio, CA, USA). Detailed information about antibodies is listed in Appendix A.

### 4.2. Identification of Differentially Expressed Genes and Overall Survival Analysis of LUAD Patients

Heat shock protein genes (Appendix A) were collected from the Enzo Life Sciences (https://www.enzolifesciences.com/platforms/proteostasis/protein-synthesis-folding/heat-shock-response/heat-shock-protein-gene-table/) accessed on 22 December 2019, and the transcriptome sequencing profiles and clinical data of LUAD patients and healthy population were downloaded from TCGA. The transcriptome levels of 88 genes were compared when the threshold was set as FDR < 0.05 and |log2FoldChange| > 1. The detailed methods of identifying the differentially expressed genes were previously reported [42]. The overall survival plots of the LUAD patients and the pathological stage plot were generated by the GEPIA [43].

### 4.3. RT-qPCR

RT-qPCR was performed as previously described [44]. All details about primers used in RT-qPCR are listed in Appendix A. The relative quantification of mRNA level was normalized to the GAPDH mRNA level with the 2^–ΔΔCt^ method. 

### 4.4. Western Blot

Western blot was performed as previously described [44]. In western bolt assays, the total protein concentrations of cell lysates were determined by BCA assays. Equal amounts of protein were subjected to the SDS-PAGE gel.

### 4.5. Cell Proliferation and Colony Formation

Cells were seeded into 96-well plates (1500 cells/well) and the proliferation was measured with CCK8 reagent (Dojindo Laboratories, Kumamoto, Japan). Cells were seeded into 6-well plates (1000 cells/well). After 14 days, colonies were stained with crystal violet and counted.

### 4.6. Migration and Invasion Assay

Cell migration and invasion abilities were determined by the transwell assay as previously [45]. In brief, cells (2 × 10^4^ cells per well) were seeded into the upper chamber of inserts (8 μm pore size, Merck Millipore, Darmstadt, Germany). Under the microscope (Olympus IX71, Tokyo, Japan), the number of migrated or invaded cells was counted. In contrast to the migration assay, we added matrigel (BD Biosciences, San Jose, CA, USA) to coat the upper chamber of inserts in the invasion assay.

### 4.7. Small Interfering RNA Transfection

Small interfering RNAs (siRNAs) targeting genes were synthesized by GenePharma (Suzhou, China). Cells were transfected with siRNA using jetPRIME Transfection Reagent (Polyplus, NY, USA) according to the standard procedure. The sequences of siRNAs are listed in Appendix A.

### 4.8. Cell Apoptosis Analysis and Cell Cycle Analysis

Apoptosis was determined by flow cytometry analysis (BD Biosciences, San Jose, CA, USA). Cells were collected and stained with propidium iodide (PI) and FITC-Annexin V (Cwbiotech, Beijing, China) according to the protocol. As for cell cycle analysis, cells were fixed with 75% cold ethanol overnight, stained with PI and analyzed with flow cytometry. The results of flow cytometry were analyzed with FlowJo (FlowJo, LLC, Ashland, OR, USA).

### 4.9. ATP Level Detection

The intracellular ATP levels were determined by the ATP detection kit (Beyotime Biotechnology, Shanghai, China) according to the manufacturer’s instructions and measured by FLUOstar Omega (BMG, Offenburg, Germany). The values were normalized to the protein quantity.

### 4.10. Immunohistochemistry

As previously described [45], immunohistochemistry and immunofluorescence (IF) were performed as previously described. The anti-Ki67 antibody was used to stain tumor tissue sections from the subcutaneous model. A lung tissue microarray derived from LUAD patients and healthy individuals was purchased from Bioaitech (Xi’an, China) and stained with an anti-CCT3 antibody. The legitimacy of tissue samples supplied by this company is provided in the supplementary material, and informed consent was obtained from the patients. Tissues were scored based on the staining intensity. 

### 4.11. Quantitative Proteomic Analysis by LC-MS/MS

The quantitative Tandem Mass Tag Mass Spectrum (TMT-MS) analysis of proteomic was performed as described previously [46]. Briefly, cells were lysed with 8 M urea containing the EDTA-free protease inhibitor cocktail (Roche, Basel, Switzerland). Proteins were digested and labeled with the TMT 10-pex reagents (Thermo Fisher Scientific, Waltham, MA, USA). Then, the labeled peptides were mixed following determination by LC-MS/MS analysis. The MS/MS spectra were explored with the UniProt human database using Proteome Discoverer software (version 2.3).

### 4.12. Transmission Electron Microscopy

The sample preparations were performed according to standard protocols for electron microscopy [47]. The samples were visualized with a transmission electron microscope (H-7650B) operated at 80 kV. The data were analyzed using NIS-elements microscope imaging software (NIS-Elements AR 3.0, Tokyo, Japan) [48]. The formula is as follows: Circularity = (4 × π × Area)/Perimeter^2^. Circularity of a perfect circle is 1.0. As the value approaches 0, the shape becomes longer.

### 4.13. Glycolysis Stress Test and Mitochondria Stress Test

The Glycolysis stress test and mitochondrial stress test were evaluated with extracellular acidification rate (ECAR) and oxygen consumption rate (OCR), respectively, with the Seahorse XFe^96^ Extracellular Flux Analyzer (Agilent Technologies, Santa Clara, CA, USA) according to the manufacturer’s instructions. The values were normalized to the protein quantity and analyzed with Wave software (Agilent Technologies, Santa Clara, CA, USA).

### 4.14. Lentivirus Infection 

Lentivirus infection was performed to silence genes stably. To aid in the packaging of lentivirus in the HEK293T cell line, the vectors pVSVG and pSPAX2 were used. The pLKO-Tet-On lentiviral vector was used to knock down CCT3 in H1299 and A549 cell lines in a doxycycline-dependent way [49,50]. The sequences of oligonucleotides used in this system are listed in Appendix A.

### 4.15. SUnSET Assay

The surface sensing of translation (SUnSET) assay was used to monitor intracellular protein synthesis [24]. Puromycin at a final concentration of 10 μg/mL was used to label newly synthesized proteins.

### 4.16. Co-Immunoprecipitation and Pulldown Assay

Cells were lysed and incubated overnight at 4 °C with anti-CCT3 or EIF3G antibodies. Then, protein A-Agarose was added and incubated (Roche, Basel, Switzerland) for 4 h at 4 °C. The complexes were analyzed by Western blot. 

For the pull-down assay, the recombinant proteins of rEIF3G and rCCT3 were covalently coupled to Cyanogen Bromide-Activated Matrices (Sigma Aldrich, St. Louis, MI, USA), respectively. Then, the complexes were blocked with glycine. After that, the complexes were incubated with rCCT3 or rEIF3G overnight at 4 °C, respectively. The precipitates were analyzed by Western blot.

### 4.17. Animal Studies

All animal studies were approved by the Institutional Animal Care and Use Committee of Tsinghua University (Approval Number: 20-LYZ3) on 27 September 2020 and were performed as previously described [51]. All animal studies were conducted on 4-week-old female BalB/C nude mice (Vital River, Beijing, China). As mentioned earlier, we constructed H1299-knockdown CCT3 cell lines with a doxycycline-inducible system. For the subcutaneous mouse model, we subcutaneously inoculated 5 × 10^6^ constructed cells into the armpit of the right limb of mice. The tumor size was monitored based on the formula volume = 0.5 × length × (width)^2^. When the tumor volume size reached 100 mm^3^, the mice were orally administered with 2 mg/mL of doxycycline in water containing 5% sucrose (dox) to induce knockdown CCT3, while the control groups were only administered with water containing 5% sucrose (vehicle). 

Then, 5 × 10^6^ cells were injected into the left pulmonary lobes of mice, and 2 × 10^6^ constructed cells were injected into mice via tail vein injection to create orthotopic and metastatic models, respectively. Then, 2 mg/mL of doxycycline in water containing 5% sucrose was administered orally to mice to induce knockdown of CCT3, and the control groups were administered vehicle. At the end of the experiments, the mice were sacrificed and the lungs were stained with H&E to identify tumor distribution.

### 4.18. Statistical Methods

Data are displayed as mean with SD, mean with SEM or median with interquartile range. Except for animal studies, the data were collected from at least three independent experiments. The Mann-Whitney test was applied to compare the gene expression levels between the LUAD and normal lung tissues. GraphPad Prism 6 Software was used for plotting and analyzing data. The distribution types of the data were checked with the Kolmogorov–Smirnov normality test. For normal distribution, a two-tailed, unpaired Student’s *t*-test was applied, and the Mann–Whitney test was performed for abnormal distribution to compare the statistical significance between the two groups. The chi-square (χ^2^) test was used to evaluate the correlation between clinicopathologic stage and protein expression levels. Kaplan–Meier curve was used to show the overall survival of LUAD patients with the log-rank (Mantel–Cox) test to determine the difference. One-way ANOVA was employed to analyze the differential gene expression in different stages of cancer cell. The *p* value less than 0.05 was defined as significant (ns, not significant; * *p* < 0.05, ** *p* < 0.01, *** *p* < 0.001, **** *p* < 0.0001).

## 5. Conclusions

In this work, we identified that CCT3 is up-regulated in LUAD and positively correlated with tumor malignancy. Patients with higher expression of CCT3 have shorter survival. Animal studies show that silencing CCT3 significantly inhibits the tumor growth and metastasis of LUAD. Proliferation and colony formation abilities are suppressed in CCT3-deficient cells. Cell cycle arrest and apoptosis are caused by knockdown of CCT3. In addition, cell migration and invasion are inhibited after ablation of CCT3 in LUAD cells. Mechanistically, CCT3 depletion inhibits glycolysis in LUAD cells, resulting in decreased intracellular ATP levels. Another point is that cytoplasmic translation is impaired in CCT3-knockdown LUAD cells by down-regulating the protein level of EIF3G. Our findings on CCT3 provide a potential therapeutic target for LUAD.

## Figures and Tables

**Figure 1 ijms-23-03983-f001:**
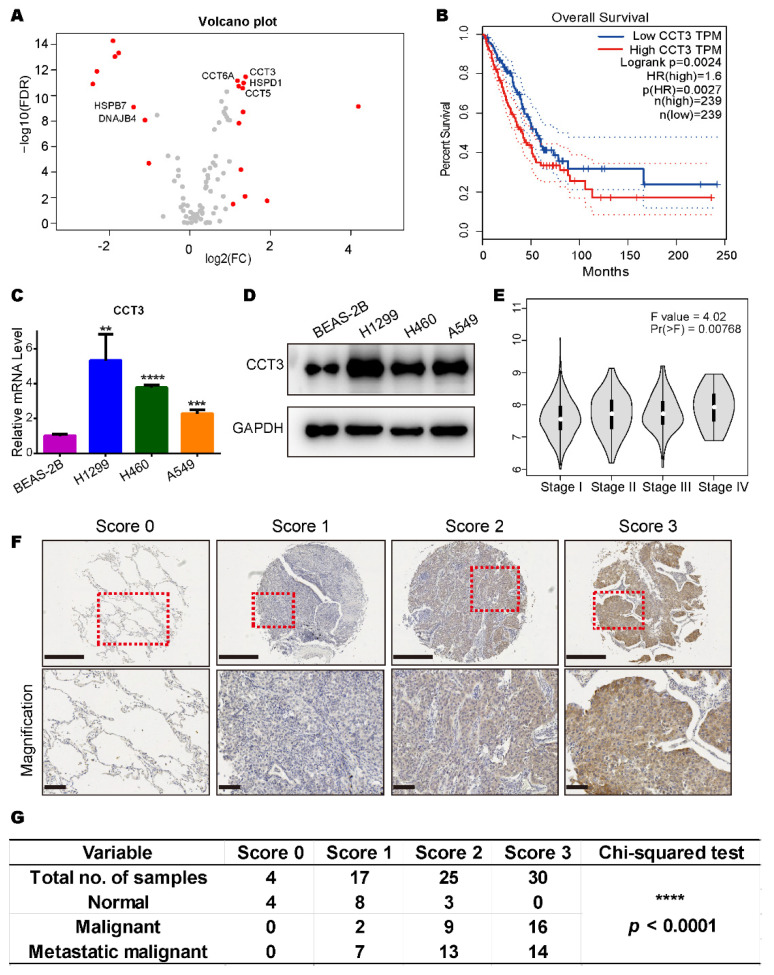
CCT3 is abundantly expressed in lung adenocarcinoma and positively correlated with tumor malignancy. (**A**) Volcano plot of HSP gene expression (535 LUAD tissues vs. 59 normal tissues). Red dots represent genes with |log2 (FC)| > 1 and FDR < 0.05. FC, fold change. FDR, false discovery rate. Raw data were downloaded from TCGA database. (**B**) Kaplan–Meier curve for overall survival of LUAD patients with high or low CCT3 expression. HR, hazard ratio. The plot was created by Gene Expression Profiling Interactive Analysis (GEPIA). Difference was assessed using the log-rank test. The dashed lines represent the 95% confidence interval. (**C**,**D**) CCT3 mRNA (**C**) and protein (**D**) levels in BEAS-2B, H1299, H460 and A549 cells. Relative mRNA levels of CCT3 were normalized to the mRNA levels of GAPDH. Representative data were from three independent experiments. Data are shown as mean + SD, ** *p* < 0.01, *** *p* < 0.001, **** *p* < 0.0001, two-tailed Student’s *t*-test. (**E**) Expression levels of CCT3 in different tumor stages of LUAD. The pathological stage plot was made by GEPIA. One-way ANOVA was employed to analyze the differences in gene expression in different stages of LUAD. (**F**) Representative images of the protein levels of CCT3 in the lung tissues with the corresponding IHC scores. Tissues were scored based on the staining intensity. The red dashed boxes represent the amplified views. Upper image scale bar = 500 μm, lower image scale bar = 100 μm. (**G**) Scores represent the IHC staining intensity of CCT3 in (**F**). 0 = negative, 1 = low positive, 2 = positive and 3 = high positive. The chi-squared test was used to evaluate the association among the categorical variables. **** *p* < 0.0001.

**Figure 2 ijms-23-03983-f002:**
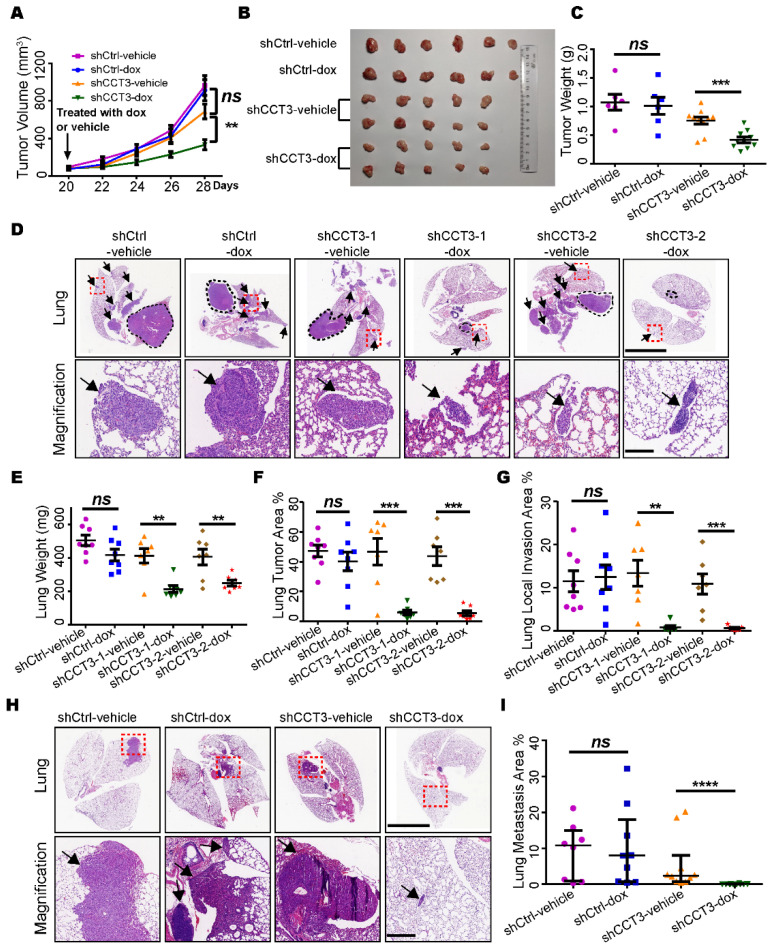
Silencing CCT3 inhibits the tumor growth and lung metastasis of LUAD. (**A**) Tumor growth in the subcutaneous tumor model. When the tumor volume reached 100 mm^3^, the mice were administrated with dox or vehicle, as indicated, to induce the knockdown of CCT3 in tumor cells. N = 6 or 10/group. (**B**) Tumors resected from each group (**A**). (**C**) Final tumor weight in each group (**A**). (**D**) Orthotopic tumor model. Representative H&E images of the lungs inoculated with different cell-line groups (n = 7 or 8/group) treated with dox or vehicle as indicated. The black dotted lines mark the primary tumors. The black arrows indicate the lung local invasion of tumor cells. The red dashed boxes indicate the amplified positions. The scale bar of upper lung images is 5 mm, and the scale bar of bottom amplified views is 200 μm. (**E**–**G**) Lung wet weight (**E**), lung tumor area (**F**) and lung local invasion area (**G**) were quantitated in the orthotopic tumor model (**D**). (**H**) Lung metastasis in mice after intravenous injection with tumor cells treated as indicated in the different groups (n ≥ 8/group). The red dashed line boxes indicate the amplified positions, the black arrows indicate the metastasis tumor. The scale bar of upper lung images is 5 mm, and the scale bar of bottom amplified views is 200 μm. (**I**) Quantification of lung metastasis areas. Dox, 2 mg/mL doxycycline in 5% sucrose water; vehicle, 5% sucrose water. Data are represented as mean ± SEM or median with interquartile range; ns, not significant; ** *p* < 0.01, *** *p* < 0.001, **** *p* < 0.0001, Mann–Whitney or two-tailed Student’s *t*-tests.

**Figure 3 ijms-23-03983-f003:**
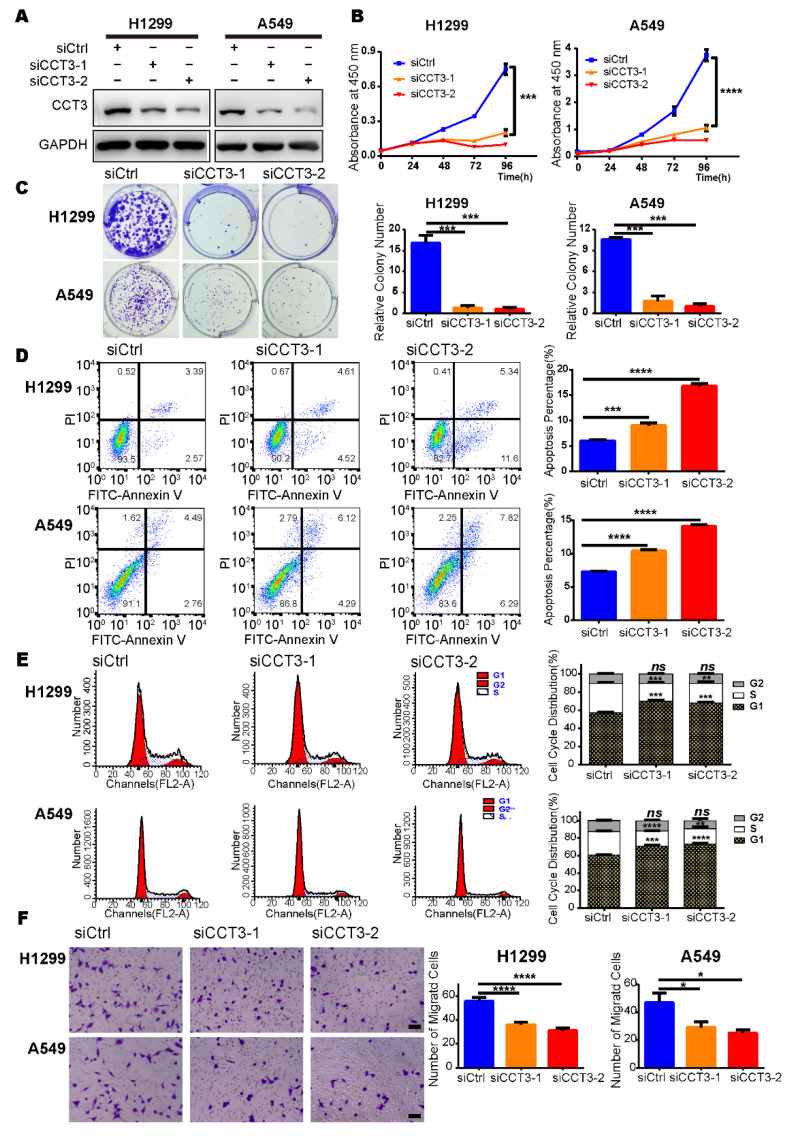
Suppression of CCT3 inhibits the growth and motility of LUAD cells. (**A**) The CCT3 knockdown efficiencies were evaluated by Western blot. (**B**) Cell proliferation (n = 5/group) and (**C**) colony formation (n = 3/group) in CCT3 knockdown cells and corresponding control cells. Representative images of colony formation assays (**left**) and relative colony numbers (**right**). (**D**) Representative images (**left**) and quantitative results (**right**) of apoptosis in LUAD cells treated for 72 h with siRNAs targeting *CCT3* or a control. (**E**) Representative images (**left**) and quantitative results (**right**) of cell cycle distribution of LUAD cells after transfection with siRNAs targeting *CCT3* or a control for 48 h. (**F**) Representative images (**left**) and quantitative results (**right**) of the transwell assays to evaluate the migration of H1299 and A549 cells with different treatments. At least five fields were quantitated in each well. Scale bar = 100 μm. Data are shown as mean + SD; ns, not significant; * *p* < 0.05, ** *p* < 0.01, *** *p* < 0.001, **** *p* < 0.0001, two-tailed Student’s *t*-test.

**Figure 4 ijms-23-03983-f004:**
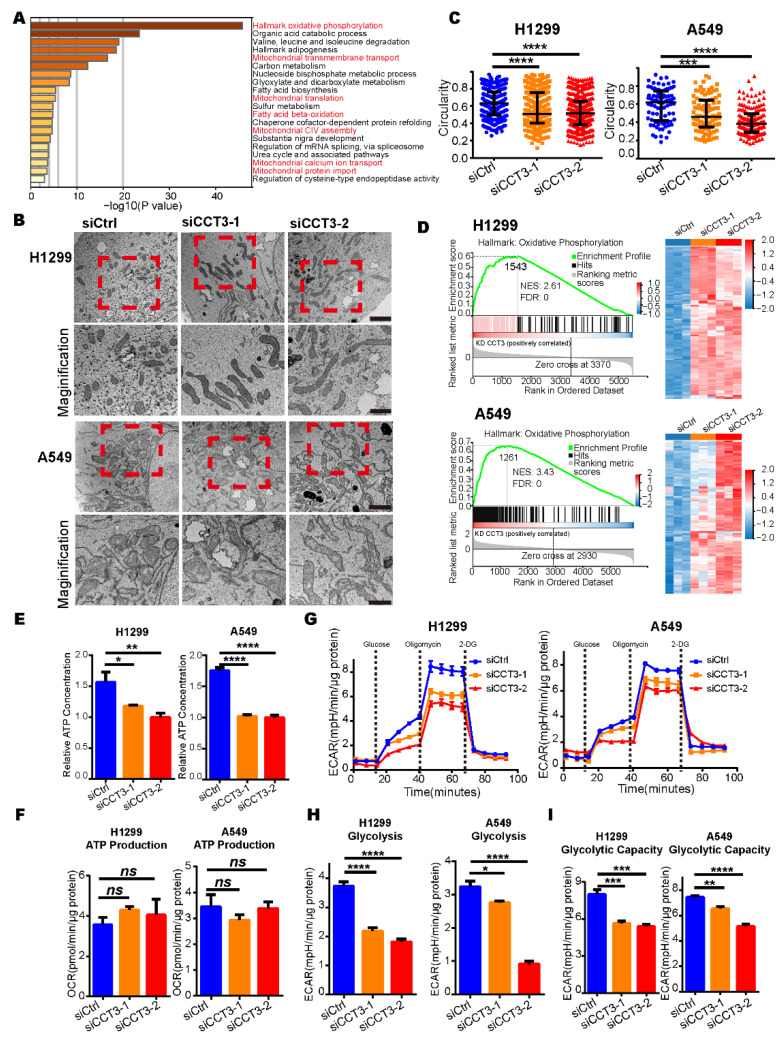
The intracellular ATP levels are decreased in CCT3-knockdown LUAD cells. (**A**) Pathway enrichment analysis of up-regulated proteins in CCT3 knockdown groups was performed with Metascape. Red font represents the pathways associated with the mitochondrion. (**B**) Representative mitochondrial images via transmission electron microscopy in H1299 and A549 cells with indicated treatments. The red dashed boxes represent the amplified visuals. Scale bar of top images = 2 μm, scale bar of bottom images = 1μm. (**C**) Quantification of the circularity of mitochondria in (**B**). Data are represented as median with interquartile range; *** *p* < 0.001, **** *p* < 0.0001, Mann–Whitney test. (**D**) GSEA enrichment plots (**left**) of the OXPHOS pathway and heat maps (**right**) of the corresponding proteins. The red represents overexpression, and the blue symbolizes down-regulation. (**E**) Intracellular ATP levels in CCT3 knockdown and control cells. The values were normalized to protein quantity. Data are represented as mean + SD, * *p* < 0.05, ** *p* < 0.01, **** *p* < 0.0001. Two-tailed Student’s *t*-test. (**F**) ATP synthesis-linked respiration used to evaluate mitochondrial function in ATP production evaluated by oxygen consumption rate (OCR). The values were normalized to protein quantity. Data are represented as mean + SEM; ns, not significant, two-tailed Student’s *t*-test. (**G**) Extracellular acidification rate (ECAR) in CCT3 knockdown and control cells measured by Seahorse analysis (glycolysis stress test). Before the experiment, the culture medium was replaced with XF assay medium supplemented with 4 mM glutamine. Discontinued lines symbolize the injections of components. The final concentrations of compounds were: 10 mM glucose, 4 μM oligomycin and 50 mM 2-deoxy-D-glucose (2-DG). (**H**,**I**) Function in basal glycolysis (**H**) and glycolytic capacity (**I**) of the cells evaluated by ECAR. Data were normalized to protein quantity and are shown as mean + SEM; * *p* < 0.05, ** *p* < 0.01, *** *p* < 0.001, **** *p* < 0.0001, two-tailed Student’s *t*-test.

**Figure 5 ijms-23-03983-f005:**
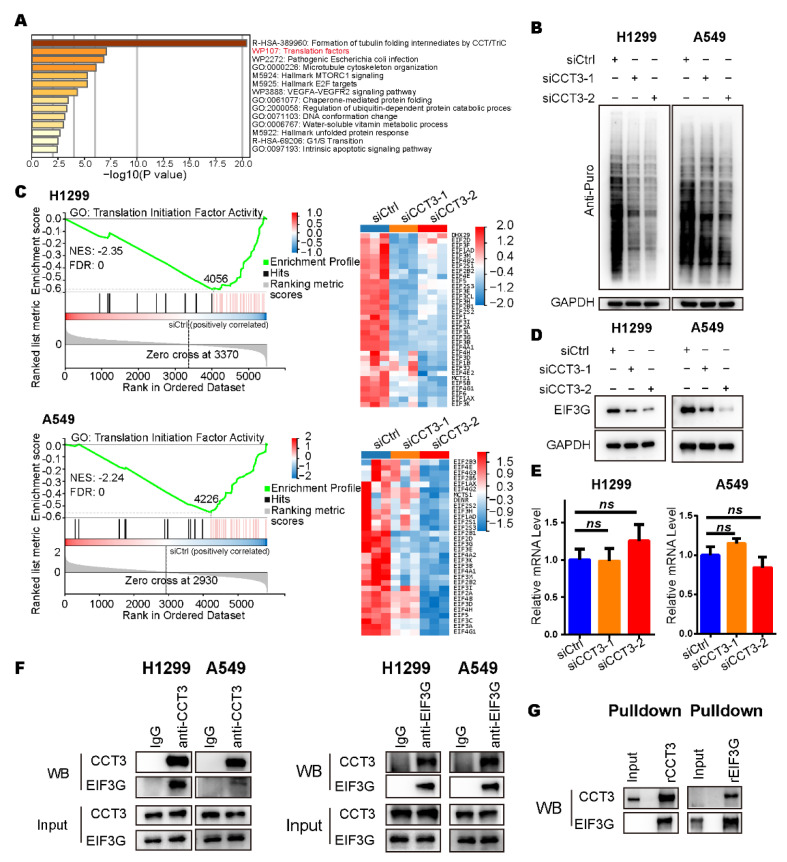
The global translation was reduced in CCT3-knockdown LUAD cells. (**A**) Pathway enrichment analysis of down-regulated proteins in CCT3-knockdown LUAD cells analyzed with Metascape. (**B**) Representative WB images of SUnSET assays in H1299 and A549 cells with the indicated treatments. Note: 10 μg/mL puromycin was added into the culture medium to label the newly synthesized peptides in cells. (**C**) GSEA enrichment plots (**left**) of the translation initiation factor activity pathway, and heat maps (**right**) of the corresponding down-regulated proteins in indicated groups. The red represents overexpression, and the blue symbolizes down-regulation. (**D**) Representative images of WB showing that EIF3G was decreased in CCT3 knockdown cells. (**E**) Relative mRNA levels of EIF3G in H1299 and A549 cells treated with siRNAs targeting *CCT3* or a control. The values were normalized to the mRNA level of GAPDH. Three independent experiments were performed. Data are shown as mean + SD; ns, not significant, two-tailed Student’s *t*-test. (**F**) Representative WB images of the interaction between CCT3 and EIF3G detected by co-IP assays. (**G**) The physical interaction between CCT3 and EIF3G was validated by pull-down assays.

**Figure 6 ijms-23-03983-f006:**
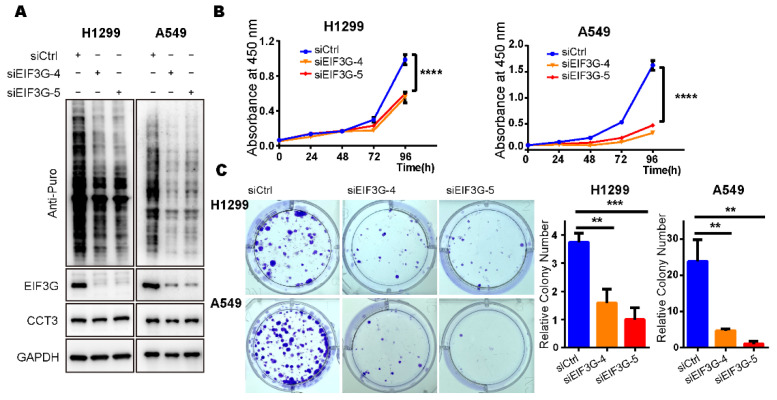
Knockdown of EIF3G alone inhibits the growth of LUAD cells. (**A**) Representative WB images of SUnSET assays in H1299 and A549 cells treated with siRNAs targeting *EIF3G* or a control. Note: 10 μg/mL puromycin was added into culture medium to label the newly synthesized peptides in cells. (**B**) Cell proliferation (n = 5/group) in indicated cells measured with CCK8 assays. (**C**) Colony formation (n = 3/group) in EIF3G knockdown cells and corresponding control cells. Representative images of colony formation assay (**left**), relative colony number (**right**). Data are shown as mean + SD, ** *p* < 0.01, *** *p* < 0.001, **** *p* < 0.0001, two-tailed Student’s *t*-test.

## Data Availability

All data generated or analyzed during this study are included in this published article and its Appendix A. The mass spectrometry proteomics data have been deposited to the ProteomeXchange Consortium (http://proteomecentral.proteomexchange.org, accessed on 25 March 2022) via the iProX partner repository [52] with the dataset identifier PXD032248. The detailed data are available from the authors.

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
