# Peer review of "Suppression of CCT3 Inhibits Tumor Progression by Impairing ATP Production and Cytoplasmic Translation in Lung Adenocarcinoma"

_ijms, 2022, doi:10.3390/ijms23073983_

Round 1

Reviewer 1 Report

In the manuscript, the authors studied the role of the CCT3 protein in lung adenocarcinoma using a wide range of molecular methods including in vitro and in vivo studies. They discovered that suppression of CCT3 inhibits lung adenocarcinoma progression by impairing ATP production and cytoplasmic translation. The methods, results, and data interpretation described in the manuscript appear to be valid, and the results will be interesting to the scientific community. However, several major and minor questions should be considered.

Major concerns:

  1. In statistical analysis the authors use chi-square and Student’s t-test. However, in most cases, the data is likely to have an abnormal distribution, and nonparametric tests (for example, the Wilcoxon test) should be used. The data distribution types should be checked with the corresponding tests for normality, and for results with abnormal distribution, nonparametric tests should be applied. Also, due to the same reason, for some figures, the median value should be denoted.
  2. The normalization to the protein content is questionable. Why the protein content is used? And what does the “protein content” mean: total amount of protein, amount of some specific protein or something else? In my opinion, the term “content” confuses. Its substitution to some other term should be considered (e.g. quantity).
  3. For evaluating ATP concentration, deproteinization should have been carried out. Then, when “protein content” was estimated?

Minor concerns:

  1. In lines 339-340, the authors discuss the value of pyruvate from glycolysis for TCA and ATP generation. However, pyruvate is also a precursor for amino acid synthesis including those which concentration were reduced in the Temiz et al. manuscript. It would be interesting to note it in the discussion.
  2. On Figure 2, “p<0.01” is repeated in the caption.
  3. The sentence “The legitimacy of tissue samples supplied by this 446 company is provided in the supplementary material, consent” seems unfinished (lines 447 and 550). Check it.
  4. In line 485, check the sentence “Then, the complexes were blocking with glycine and incubated with rCCT3 485 or rEIF3G overnight at 4ºC” for English grammar.
  5. Replace “dox” with full word form in the sentence “Then, dox was administered orally to mice to induce knock- 501 down of CCT3, while the control groups were administered vehicle” in line 501.

Reviewer 2 Report

MDPI

Overview

CCT3 is a member of heat shock protein family with a well-defined role in promoting malignant phenotype in several types of cancer. This study has addressed the role of CCT3 as a tumor-promoting gene in lung cancer. Using RNAi-mediated knock down of CCT3 as a model authors showed that silencing of CCT3 has an anti-tumor effect both in cellular and animal models. Authors defined several novel molecular mechanisms that might be responsible for the effects of CCT3-knockdown on the malignant phenotype including inhibition of ATP production and cytoplasmic translation. Moreover, EIF3G was identified as a binding partner of CCT3.

Commentaries for authors

  1. Authors define lung cancer cell lines used in the study as LUAD (lung adenocarcinoma). But, in fact, only A549 is a lung adenocarcinoma, while NCI-H1299 and NCI-H460 are large cell carcinomas, not adenocarcinomas.
  2. Two recent articles are not cited neither discussed in the manuscript: Toxicol Appl Pharmacol 2022 Mar 15;439:115926. doi: 10.1016/j.taap.2022.115926 and Bioengineered 2021 Dec;12(1):7335-7347. doi: 10.1080/21655979.2021.1971030. In these articles it was shown that (1) CCT3 is up-regulated in NSCLCL and the upregulation might be responsible for cisplatin resistance in lung cancer cells; (2) using CCT3 knockdown it was shown that CCT3 silencing induces apoptosis and cell cycle arrest and inhibits proliferation and migration of lung cancer cells. Therefore, these articles in part compromise the novelty of the presented findings. These articles should be cited and thoroughly discussed. The novelty of the findings should be revised.
  3. CCT3 knockdown results in the reduction of EIF3G protein level and CCT3 was shown to directly interact with EIF3G. Results in Fig6 show that knockdown of EIF3G leads to similar effects on cell proliferation, colony formation and protein translation as knockdown of CCT3. Based on these data authors conclude that effects of CCT3 knockdown on lung cancer cells could be at least in part mediated via EIF3G depletion. In my opinion, the presented data do not allow to draw such conclusion. Authors should preform additional experiments to prove that effects of CCT3 knockdown are indeed mediated by EIF3G. For example, authors could investigate if CCT3 knockdown phenotype is rescued by EIF3G overexpression. In addition, the effects of double knockdown of CCT3 and EIF3G could be studied (but it could be too toxic for cells, I guess).
  4. In the Discussion section (lines 317-327) authors discuss that CCT3 depletion induces decrease in microtubule proteins, and conclude that CCT3 depletion inhibits cell motility via downregulation of tubulins. The role of tubulin and MT cytoskeleton was not investigated in the present work and lies out of the scope of the manuscript. No experimental data were presented to support the role of MT protein depletion as one the mechanisms that mediate CCT3-dependent inhibition of lung cancer cell migration and cell cycle arrest.

Minor points

Line 20 – “ Mechanismly” – This word should be changed.

Fig3C -Graphs should be labeled H1299/A549 for clarity.

Authors should provide details on mitochondria circularity quantification. These data are absent from the Materials and methods section.

In Materials and methods Western blotting and RT-qPCR are combined into one paragraph which is quite unusual.

Round 2

Reviewer 1 Report

The authors took into account all the comments